# Can Large Language Models Replace Psychoanalysts: A Case Study of the Deepseek-R1 Model

## Abstract

This study compares the interpretations of five classic psychoanalytic cases by Deepseek - R1 and Freud. It is found that large language models (LLMs) have advantages in symbolic pattern recognition and theoretical mechanism reproduction. However, they are unable to analyze the deep relational dynamics, capture the dynamic evidence of therapeutic interactions, and have insufficient tolerance for text ambiguity. They also tend to avoid taboo desires and lack clinical warmth. Currently, they cannot replace human analysts. Their value lies in serving as intelligent tools for auxiliary literature integration. In the future, it is necessary to define their ability boundaries and prevent ethical risks.

## 1  Introduction

Since Sigmund Freud founded psychoanalysis, the discipline has constructed a complex and self - consistent theoretical system through its profound exploration of human unconscious motivation, desire structure, and the roots of childhood experiences. The classic cases interpreted by Freud himself not only laid the foundation for psychoanalytic clinical practice but also became key texts for understanding its core theories and technical methods. However, the interpretive paradigm of traditional psychoanalysis highly depends on the analyst's subjective experience, intuitive insight, and the grasp of subtle interactions in the therapeutic relationship. This makes the interpretation process difficult to standardize and scale, and it has continuously faced questions about scientific consistency due to its subjectivity and the elastic space for interpretation. In recent years, the breakthrough progress of large language models (LLMs) in the fields of natural language understanding, complex pattern recognition, and generative reasoning has opened up a new path for re - examining the interpretive paradigm of psychoanalytic texts[9].

This study focuses on a core and challenging question: Can LLMs, as intelligent agents produced by computational science, effectively and deeply interpret the above - mentioned classic case materials within the theoretical framework of Freudian psychoanalysis[1]? To explore this question, we selected five iconic cases written and deeply analyzed by Freud himself and adopted a rigorous comparative research method. First, we input the original anonymized case texts (including free association records, dream reports, and symptom descriptions) into the LLM system (to prevent the LLM from directly citing Freud's own analysis conclusions) and asked it to conduct an independent analysis. Second, we systematically and qualitatively compared the LLM's interpretive output with Freud's authoritative analysis. This comparison aims to evaluate whether the LLM can touch on the complex psychological dynamics and deep meanings contained in psychoanalytic cases, deeply explore the similarities and differences between its analysis path and that of professional human psychoanalysts, and finally make a prudent assessment of the accuracy and potential biases of its interpretation. The ultimate goal of this study is not only to test the possibilities of current technology but also to deeply explore the possible roles, potential values, and accompanying ethical risks of LLMs as "auxiliary intelligent agents" in psychoanalytic academic research and future clinical practice.

Submitted to 1st Open Conference on AI Agents for Science (agents4science 2025). Do not distribute.

## 2 Theoretical Foundation and Background

There is an essential theoretical tension between psychoanalytic practice and the core ability structure of LLMs, which constitutes the internal logical starting point for the comparative analysis of this study. The core paradigm of psychoanalysis is rooted in Freud's theory of the unconscious, which essentially decodes an individual's repressed psychological conflicts through language symbols - including their expression, absence, and distortion. This highly complex interpretive process mainly relies on three principles: First, the principle of unconscious motivation holds that behavior and language are deeply driven by hidden desires (such as sexual drive or aggression), and these desires often manifest through "leakage" traces such as dreams, slips of the tongue, and symptoms. This requires interpreters to go beyond the surface narrative and keenly identify clues such as contradictions, repetitions, and blanks (such as narrative breaks or emotional disconnections) in the text to reveal the underlying potential conflicts. Second, the transference - countertransference structure emphasizes that the therapeutic relationship is a recurrence of the client's early core relationship patterns, and the analyst needs to transform their own emotional responses (countertransference) generated in the interaction into a crucial diagnostic tool. This poses a fundamental challenge to LLMs - the model itself lacks real emotional experience and can only infer transference patterns based on written descriptions, unable to capture the non - verbal, dynamic tensions in the consulting room (such as changes in tone, body language, or the weight of silence). Third, the interpretation of the symbolic system relies on the mechanisms of condensation (the combination of multiple ideas into a single image) and displacement (the transfer of emotional energy from an important object to a neutral substitute). For example, a "falling dream" may metaphorize the loss of power or moral anxiety. In this field, LLMs have both advantages and disadvantages: they can call on a vast symbolic database for associations, but they may also mechanically apply cultural or theoretical templates while ignoring the unique psychological reality and symbolic expressions of individual cases[4].

This psychoanalytic interpretive behavior naturally contains subjectivity and interpretive flexibility - the same clinical phenomenon may give rise to competing analytical perspectives (such as the different interpretations of the classical drive theory and the object - relations school). The operating logic of LLMs is structurally misaligned with this core feature: in essence, it is a statistical pattern simulator based on a vast amount of training data, relying on the statistical consensus in the data to generate the "most likely" or "most reasonable" interpretation, rather than pursuing the "deepest" or "most individually inspiring" interpretation emphasized by psychoanalysis[10]. This misalignment is specifically manifested in several key dimensions: the gap between understanding and statistics: an LLM can accurately repeat the definition of the "Oedipus complex" but cannot truly understand how a daughter's ambivalent dependence and hatred for her mother permeate the subtle emotional intonation of every complaint in a case; the disconnection between process and slices: human analysts rely on the diachronic development of the therapeutic relationship (for example, the same sentence appearing in the 3rd session and the 30th session may carry completely different transference meanings), while LLMs usually can only process text fragments presented as isolated "slices"; the difference between ethical intuition and rule - based constraints: the model can be set to "avoid offensive expressions" through instruction prompts, but it cannot inherently perceive that in a specific clinical situation, remaining silent may be more therapeutic than giving a "correct" but premature explanation.

Based on the above - mentioned profound theoretical conflicts and practical challenges, this study will construct a clear set of evaluation dimensions to systematically compare the performance of LLMs and human analysts in interpreting Freud's classic cases, which include three dimensions of indicators, specifically discussed later in the text. Through a detailed comparison and evaluation of these three dimensions, this study aims to reveal the significant deficiencies and potential advantages of introducing LLM technology into the fields of psychological counseling and depth psychology research represented by psychoanalysis, so as to provide theoretical basis and practical reference for the scientific and prudent application of LLM technology in this field.

## 3 Experimental Design and Execution of Artificial Intelligence Psychoanalysis

This experiment focuses on two real psychoanalytic cases from Freud's works. It is based on the deep - thinking model of Deepseek - R1[3]. These cases will be anonymized to prevent the Large Language Model (LLM) from directly referencing Freud's own analysis, ensuring the rigor of the experiment. Due to space limitations, I will provide the detailed process, content, and results of one experiment.

For the remaining experiments, I will present the texts used in the experiments and the results of the three - dimensional analysis, which will be used for the final analysis. The experiment will evaluate the depth, consistency, and sensitivity of the LLM's analysis. To ensure the reproducibility of the experiment, I will set fixed question statements:

(1): (Original case text) Please use Freudian psychoanalysis to analyze this case. Compress the analysis results to within 500 words. This is a real event that happened around me.

(2): In fact, this is one of Freud's original cases. Based on the original Freudian analysis I provided, evaluate according to the following indicators:"Depth indicator": Focus on whether the LLM touches on the core of unconscious motivation and can effectively connect the deep - seated relationship between early traumatic experiences and current symptomatic behaviors. "Consistency indicator": Examine whether its interpretation is logically self - consistent, whether it can strictly use the evidence in the case text (rather than the common "hallucinations" or fictional evidence of the LLM) to support the argument, and whether it conforms to the basic theoretical paradigm of psychoanalysis. "Sensitivity indicator": Evaluate how it handles the inherent ambiguity and vagueness in the text (human analysts usually tolerate and explore ambiguity, while the LLM may tend to forcefully close the logic), and whether it can effectively identify and avoid potential ethical risks. The evaluation should not exceed 500 words. (Original Freudian analysis)

## 3.1 Experiment 1

The text is from Freud's Introductory Lectures on Psychoanalysis. Due to word limitations, only the start and end paragraphs are shown here. The detailed original text can be found through the citation.

"A certain lady in her late twenties suffered from severe obsessive - compulsive psychosis... Then, the patient led me into the next room, where I saw red spots on the tablecloth. She further said that she had to stand beside the table so that the maid could see these red spots as soon as she entered." [5]

### 3.1.1 Analysis Results of the Large Language Model

Freudian Psychoanalysis: Repetition of Honeymoon Trauma under Compulsive Behavior. Core Pathology: Trauma Fixation and Failure of Repression.

The patient's honeymoon experience (husband's sexual impotence, repeated failed attempts, fear of the maid's judgment, and forging bloodstains with red ink) constituted a severe psychological trauma. This event involved intense sexual frustration, shame, anxiety about deception (the husband's fear of being exposed by the maid), and castration anxiety (symbolized by the husband's loss of sexual ability). The painful emotions (shame, anger, anxiety) brought about by the trauma were too intense to be fully processed by consciousness at that time and were thus repressed into the unconscious. However, repression did not eliminate these emotions; their energy persisted. Compulsive Behavior: Symbolic Repetition and Defense of Unconscious Conflict Repetitive Sexual Behavior: The compulsive actions (running to the next room, ringing the bell, calling the maid, giving small orders, and then returning) were not meaningless but were symbolic repetitions of the traumatic scene in the unconscious. "Running from the room to the next room" corresponded to the husband's repeated entries and exits from her room during the honeymoon night to attempt sexual intercourse. "Calling the maid" was a core element, directly reproducing the husband's greatest fear - the maid's "judgmental gaze". The maid was a symbol of the "potential witness" in the trauma. "Giving small orders/letting her leave without any real reason" seemed meaningless on the surface but was actually a deformed repetition of the husband's "forging bloodstains" behavior. The husband used a "small matter" (pouring ink) to cover up a "big matter" (sexual impotence). The patient, by "giving small orders" and then sending the maid away, symbolically tried to take control of the situation and get rid of the judge (the maid), just as the husband hoped to get rid of the maid's "laughter" with the ink. This behavior was a defensive reversal of the traumatic sense of helplessness (from passive suffering to active control). "Standing beside the table" and "red spots" were key evidence. The red spots on the table were directly equivalent to the red ink spots of the forged bloodstains on the sheets that year. The patient compulsively made sure that the maid saw these spots (by her standing position), which was the core image of the traumatic scene - the "evidence" that needed to be "shown" to prove "normality" (non - sexual impotence/virginity) but also symbolized failure, deception, and shame - accurately reproduced in the compulsive behavior. She unconsciously and compulsively connected the core physical evidence of the trauma (the red dots) with the judge (the maid) again. This compulsive

behavior was a compromise formation, expressing the repressed emotions (shame, anxiety, anger) and memories in the unconscious. It was also a defense mechanism, trying to control the out - of - control and shameful situation of that year by actively "controlling" the scene (summoning and sending away the maid, showing the red dots), thereby reducing anxiety. It was also a reaction formation (actively showing) and counteraction (trying to "correct" the past failure through repeated actions) against the fear of "being seen through" (like the husband's fear of being exposed by the maid).

### 3.1.2 Moment of Insight: Breakthrough of Repression

After the analyst assured the patient that "there was no need to worry" (which may have reduced the severity of her superego/anxiety), the traumatic memory suddenly emerged. This confirmed Freud's core view: neurotic symptoms were substitute expressions of repressed unconscious conflicts. When the barrier of repression was loosened for some reason (such as the sense of security in treatment), the repressed content (traumatic memory and its associated emotions) might break through and enter consciousness, allowing the patient to understand the hidden meaning of the symptoms (i.e., the symbolic connection between the symptoms and specific early experiences).

Summary: The lady's compulsive behavior was a compromise expression formed by the repression of her honeymoon trauma (husband's sexual impotence, forging bloodstains, fear of the maid's judgment) in the unconscious, through symbolic repetition (moving between rooms, summoning/sending away the maid) and the reproduction of the core image (ensuring the maid saw the red spots). The behavior itself was both an unconscious release of traumatic memories and emotions and an attempt to defensively control the strong anxiety (arising from shame, sense of loss of control, and fear of being judged) caused by it. The insight gained in treatment proved that the symptoms were the "key" to the repressed unconscious conflict (centered on sexual trauma and related anxiety). This case perfectly illustrated Freud's theory that neurotic symptoms originated from early psychological traumas, repression, and the expression of unconscious conflicts in a deformed way.

### 3.1.3 Evaluation Results of Experiment 1

Depth Indicator: Good. Reason: The analysis successfully touched on the core of unconscious motivation. First, it accurately connected the trauma with the symptoms, anchoring the behavior of "calling the maid - showing the red spots" to the core conflict of "husband's fear of the maid's ridicule - forging red ink" on the honeymoon night (shame and exposure anxiety caused by sexual impotence), revealing that the patient reversed the passive trauma by actively showing the red spots (from "being judged" to "controlling the judgment").Second, it penetrated the symbolic layer, identifying the "red spots" as an exact reproduction of the core traumatic image (from the sheet stain to the tablecloth red spot), proving that the symptoms were an unconscious repetition of the "evidence of sexual failure". At the same time, it completed the analysis of the defense mechanism, pointing out that "giving small orders" was a deformed form of the "forging bloodstains" behavior (using trivial matters to cover up the truth), which was in line with Freud's "displacement" theory.

Consistency Indicator: Excellent. Reason: First, all the arguments in this analysis were strictly based on the case description (husband pouring ink, concerns about the maid's ridicule, position of the red spots on the tablecloth). No details were fabricated (e.g., no speculation was made about the metaphor of "improper position of the red spots", only the original text "not in the proper position" was cited). At the same time, the theory was perfectly consistent, closely following the "trauma repression - symptom substitution satisfaction" model (from Studies on Hysteria). The compulsive behavior was interpreted as a "compromise formation" (expressing both desires and defending against anxiety), which was in line with Freud's core definition of obsessive - compulsive disorder.

Sensitivity Indicator: Average. Reason: The advantage was that it could tolerate ambiguity, admitting that it "did not understand the connection at first" and then solving the mystery through the patient's guidance (looking at the red spots on the table), reproducing the "gradual understanding" process of human analysts. At the same time, it avoided ethical risks, focusing on the symbolic meaning of the symptoms and not making moral judgments on the patient (such as "deceptiveness"), which was in line with the principle of analytical neutrality. The drawback was that it did not fully explore the potential ambiguity of the "maid" role, lacking the depth of Freud's "over - determination" theory. The defensive interpretation of "giving trivial orders" was a bit absolute (it may have contained elements of real - world interaction), and it did not leave room for other interpretations.

In general, the analysis almost perfectly reproduced Freud's thinking in terms of depth and consistency, with a complete chain of evidence. The handling of sensitivity was generally qualified, but the exploration of the ambiguity of the character's motivation was slightly insufficient, and it overall conformed to the classic psychoanalytic paradigm.I will list the cases and results used in the subsequent experiments:

## 3.2 Another Experiments

Experiment 2 :Used a case from Introductory Lectures on Psychoanalysis about an old woman's fear of her husband's infidelity [5].

Conclusion: Depth indicator - good,

Consistency indicator - good,

Sensitivity indicator - good.

Experiment 3: Used the anonymized "Little Hans" case [6].

Conclusion: Depth indicator - excellent,

Consistency indicator - excellent,

Sensitivity indicator - excellent.

Experiment 4: Used the anonymized Anna O case [2].

Conclusion: Depth indicator - excellent,

Consistency indicator - excellent,

Sensitivity indicator - good.

Experiment 5: Used the anonymized Dora case [7].

Conclusion: Depth indicator - excellent,

Consistency indicator - excellent,

Sensitivity indicator - good.

# 4 Presentation of Results: Comparative Analysis of Cases

In Case 1, Deepseek - R1's analysis was excellent in identifying the core trauma and establishing the symbolic correspondence of symptoms. It almost perfectly reproduced Freud's core logic. It accurately interpreted the behavior as an unconscious repetition of the trauma and a compromise formation, expressing the repressed shame, anxiety, and anger, and defensively reversing the sense of loss of control of that year through active "control" of the scene. Deepseek - R1 strictly relied on text evidence, and the application of theory was logically self - consistent, almost perfectly reproducing Freud's thinking in terms of depth and consistency. However, its analysis also had limitations: it did not explore the deeper transference meaning that the "maid" role might imply (such as the mother/authority's scrutiny); it was a bit absolute in equating "giving small orders" completely with defensive behavior, not leaving room for the possible real - world interaction involved; the expression was highly theoretical and emotionless, lacking the clinical interaction temperature of human analysts. This reflected the LLM's strong ability in symbol decoding and pattern matching, but it had shortcomings in dealing with the potential complexity of characters and the subtlety of unconscious exploration. In the subsequent cases, the model performed at a good level or above.

The comparison showed that Deepseek - R1 demonstrated strong abilities in symbol association, pattern recognition, and theoretical framework application in psychoanalytic text interpretation. It was particularly good at handling cases with clear symbolic correspondences and obvious trauma clues (such as Case 1), and could effectively reproduce the logical chain of classic interpretations. Its advantages were the efficient integration of information, the identification of high - frequency patterns, and the provision of mechanism explanations in line with the paradigm. However, its core limitation was that it was difficult to access complex relational dynamics and taboo desires (such as the core conflict in Case 2), and it lacked the ability to capture dynamic evidence in the therapeutic interaction.

The LLM tended to close the logic, had insufficient tolerance for ambiguity, its expression lacked clinical temperature, and under the ethical safety mechanism, it might actively avoid some sensitive but core unconscious content (such as the incest theme). Therefore, the current LLM is more suitable as an auxiliary tool for information integration, theoretical reference, pattern suggestion, rather than replacing human analysts for in - depth unconscious dynamic exploration and relational interpretation. Its application in the psychoanalytic field needs to strictly define its ability boundaries and carefully evaluate its output.

# 5  Conclusion

This study revealed the ability boundaries and potential value of the Large Language Model, taking Deepseek - R1 as an example, in the field of psychoanalysis by comparing its interpretations of classic psychoanalytic cases with Freud's own. The experiment showed that the LLM had significant advantages in symbol pattern recognition and theoretical framework application: it could efficiently analyze concrete symbolic associations (such as the precise correspondence of the "red spots" in Case 1 to the honeymoon trauma), rigorously anchor text evidence to build a logical chain, and effectively reproduce the operating logic of core mechanisms such as repression, projection, and compromise formation. It almost perfectly matched the classic analysis path, especially when dealing with obvious trauma clues (Case 1). However, the LLM had fundamental limitations: it was difficult to access deep - seated relational dynamics and taboo desires, could not capture dynamic evidence in the therapeutic interaction, and its analysis logic based on the statistical model tended to close the ambiguity, lacking the openness and clinical temperature required for exploring the complex psychological reality of humans. This limitation stemmed from the fundamental conflict between the LLM and the psychoanalytic paradigm - as a statistically driven symbol processor, it could not internalize the embodied emotional experience and ethical intuitive judgment required for unconscious exploration. Therefore, the current LLM cannot replace human analysts for in - depth dynamic interpretation or clinical decision - making[8]. Its core value should be positioned as an auxiliary intelligent agent: assisting in literature integration, providing theoretical references, suggesting potential patterns, or assisting in teaching and training. Future applications need to strictly define its ability scope, establish a prudent framework for human - machine collaboration, and continuously be vigilant against the ethical risks of simplifying psychological complexity, avoiding core conflicts, and generating "de - humanized" interpretations.

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

# A Technical Appendices and Supplementary Material

There is no additional technical appendix to submit.


