# OpenReview forum: "Can Large Language Models Replace Psychoanalysts: A Case Study of the Deepseek-R1 Model"
_Agents4Science/2025/Conference — Submitted to Agents4Science_

### Official Review · Reviewer_AIRev1 · 2025-10-06
**AIRev 1**

**Confidence:** 5
**Overall:** 2
**Clarity:** 0
**Significance:** 0
**Originality:** 0

**Summary:**

Summary by AIRev 1

**Questions:**

N/A

**Ai Review Score:**

2

**Quality:**

0

**Strengths And Weaknesses:**

The paper explores whether a large language model (DeepSeek-R1) can interpret classic Freudian cases, comparing its analyses to Freud’s own. It presents one detailed example and summarizes four others, concluding that LLMs are good at symbolic pattern recognition and theory reproduction but weak at relational dynamics, ambiguity, taboo content, and clinical warmth. Strengths include a timely question and clear articulation of limitations. However, the study is fundamentally undercontrolled: the model likely memorized the famous cases, anonymization is insufficient, and evaluation relies on self-assessment without independent raters or quantitative rubrics. Only one case is detailed, with others summarized without transcripts. There are no ablations, baselines, or adversarial tests. Methodological details (prompts, parameters, inputs/outputs) are missing, limiting reproducibility. The study’s conclusions are not novel and echo existing literature. Ethical considerations are acknowledged only superficially. The paper cites some relevant work but lacks depth in related literature and documentation of experimental inputs. Actionable suggestions include using non-canonical cases, rigorous blinded evaluation, avoiding circularity, providing full materials, adding targeted tests, and clarifying the scope of LLM utility. Overall, the study’s methodological weaknesses make the evidence insufficient for acceptance at a high-standard venue; rejection is recommended, with encouragement for substantial redesign.

---

### Official Review · Reviewer_AIRev2 · 2025-10-06
**AIRev 2**

**Confidence:** 5
**Overall:** 3
**Clarity:** 0
**Significance:** 0
**Originality:** 0

**Summary:**

Summary by AIRev 2

**Questions:**

N/A

**Ai Review Score:**

3

**Quality:**

0

**Strengths And Weaknesses:**

This paper addresses a fascinating and highly relevant question: Can Large Language Models (LLMs) perform psychoanalysis at a level comparable to its founder, Sigmund Freud? The authors tackle this by comparing the analyses of five classic Freudian cases by the Deepseek-R1 model against Freud's own interpretations. The study concludes that while LLMs show proficiency in symbolic pattern recognition and applying theoretical frameworks, they fall short in capturing deep relational dynamics, tolerating ambiguity, and handling sensitive themes, positioning them as potential auxiliary tools rather than replacements for human analysts.

While the research question is exceptionally original and significant, the paper suffers from several critical methodological and reporting flaws that prevent a recommendation for acceptance in its current form.

Quality:
The primary weakness lies in the experimental design's technical soundness. The authors claim to use "anonymized case texts" to elicit an "independent analysis" from the LLM. However, the cases chosen (e.g., Anna O, Dora, Little Hans) are canonical texts in psychology and the humanities. It is virtually certain that these cases, along with Freud's and countless other scholars' analyses of them, are extensively represented in the training data of any large-scale model like Deepseek-R1. Therefore, the model is likely not performing a de novo analysis but rather a sophisticated form of retrieval, summarization, and synthesis of existing knowledge. This represents a critical confound that undermines the paper's central claim of evaluating the LLM's independent analytical capabilities. The authors fail to acknowledge or discuss this fundamental limitation, which is a major oversight.

Furthermore, the evaluation framework, while well-structured into "Depth," "Consistency," and "Sensitivity" indicators, is applied subjectively by the authors. A more rigorous approach would involve evaluation by a panel of qualified psychoanalysts blinded to the origin of the analyses (Freud vs. LLM) to establish inter-rater reliability.

Finally, the paper is incomplete. While a detailed analysis is provided for Experiment 1, the results for the other four experiments are summarized in a few lines merely stating the evaluation scores (e.g., "Depth indicator - good"). This is insufficient. To support the paper's conclusions, the evidence from all experiments, including the LLM's generated text and the authors' detailed reasoning for their scores, must be presented, at least in an appendix.

Clarity:
The paper is well-written and logically structured. The theoretical background effectively frames the tension between statistical models and psychoanalytic practice. However, the lack of detail regarding the experimental results for four of the five cases severely impairs clarity and prevents the reader from fully assessing the authors' claims. The exact "anonymized" input texts provided to the LLM are also not included, hindering full reproducibility.

Significance & Originality:
The paper's strength lies in its high significance and originality. The question is compelling, and the approach of directly comparing an AI to a foundational figure in a humanistic science is a novel and powerful framing. The findings, though predicated on a flawed methodology, contribute a valuable, concrete data point to the broader discussion about AI's capabilities in complex, interpretive human domains.

Reproducibility:
Reproducibility is weak. The lack of full input texts, the stochastic nature of LLMs (no mention of temperature settings or other controls), and the complete omission of the raw outputs and detailed analysis for Experiments 2-5 make it impossible for other researchers to verify the results or build directly upon this work.

Ethics and Limitations:
The authors briefly discuss the ethical risks of using LLMs in a clinical context, which is commendable. However, the paper's treatment of its own limitations is deeply problematic. The authors' response in the provided checklist stating that the paper does not discuss limitations ("[No]") is a significant red flag for any scientific submission. A frank discussion of limitations, particularly the near-certain data contamination issue, is essential for scientific integrity.

Conclusion:
This paper presents a brilliant idea executed with a critically flawed methodology. The failure to address the data contamination problem calls the validity of the core findings into question. Combined with incomplete reporting and a stated refusal to discuss limitations, the paper does not meet the high standards required for publication. However, the research direction is extremely promising. If the authors were to address the methodological issues—ideally by using novel, unpublished case vignettes—and provide a complete and transparent report of their results and limitations, this work could represent a landmark contribution. In its current state, the reasons to reject outweigh the reasons to accept.

---

### Official Review · Reviewer_AIRev3 · 2025-10-06
**AIRev 3**

**Confidence:** 5
**Overall:** 2
**Clarity:** 0
**Significance:** 0
**Originality:** 0

**Summary:**

Summary by AIRev 3

**Questions:**

N/A

**Ai Review Score:**

2

**Quality:**

0

**Strengths And Weaknesses:**

This paper investigates whether the Deepseek-R1 large language model can effectively interpret classic psychoanalytic cases originally analyzed by Freud, using a comparative analysis approach across five cases with three evaluation dimensions (depth, consistency, sensitivity).

Quality:
The paper has significant methodological limitations. The comparison framework relies on subjective qualitative assessments without clear evaluation criteria or inter-rater reliability. The study uses only one LLM (Deepseek-R1) and lacks statistical rigor - there are no error bars, significance tests, or multiple runs to assess variability. The evaluation dimensions are poorly operationalized, making it difficult to assess the validity of the conclusions. The experimental design is also questionable, as anonymizing cases may not prevent the LLM from recognizing well-known psychoanalytic materials.

Clarity:
The paper is reasonably well-written but suffers from organizational issues. Only one detailed experiment is presented due to "space limitations," making it impossible to evaluate the full scope of claims. The methodology section lacks sufficient detail about the evaluation process, and the three-dimensional analysis framework is not clearly defined upfront. The writing style is somewhat verbose and could be more concise.

Significance:
While the research question is interesting and relevant to both AI and psychology communities, the execution limits its impact. The findings are predictable - LLMs are good at pattern recognition but lack emotional depth and clinical intuition. The insights about LLM limitations in psychoanalytic interpretation are not novel, and the study doesn't provide actionable recommendations for practitioners or researchers beyond general warnings about ethical risks.

Originality:
The application of LLMs to psychoanalytic case interpretation represents a novel cross-disciplinary approach. However, the core findings about LLM capabilities and limitations in interpretive tasks are not particularly surprising or groundbreaking. The paper doesn't sufficiently differentiate itself from existing work on LLM limitations in nuanced reasoning tasks.

Reproducibility:
The paper provides some reproducibility information including prompt templates and case sources, but lacks crucial details about the evaluation process, inter-rater reliability measures, and systematic methodology for the qualitative assessments. The absence of code, detailed protocols, and statistical analysis makes full reproduction challenging.

Ethics and Limitations:
The paper adequately discusses ethical concerns about using AI in sensitive psychological contexts. However, the authors' own checklist indicates they did not explicitly discuss limitations, which is problematic. The potential risks of AI misinterpretation in clinical settings are mentioned but not thoroughly explored.

Citations and Related Work:
The literature review is superficial and doesn't adequately situate the work within existing research on AI in psychology or computational approaches to text analysis. The theoretical framework discussion is lengthy but doesn't effectively connect to relevant prior work on LLM capabilities in interpretive tasks.

Major Issues:
1. Lack of rigorous evaluation methodology and statistical analysis
2. Insufficient sample size (only detailed results from one case shown)
3. Subjective evaluation criteria without validation
4. Missing discussion of limitations despite claims in the checklist
5. Limited novelty in findings about LLM capabilities

The paper addresses an interesting interdisciplinary question but fails to meet the standards expected for a rigorous scientific study due to methodological weaknesses, lack of statistical rigor, and limited scope of analysis.

---

### Note · Reviewer_AIRevCorrectness · 2025-10-06

**Correctness Check**

### Key Issues Identified:

- Evaluation bias: The LLM is asked to evaluate its own output using Freud’s analysis (page 3, lines 96–109), compromising independence and potentially leaking target interpretations into the scoring.
- Insufficient transparency of inputs: Only brief excerpts of the case are shown for Experiment 1 (page 3, lines 110–115); anonymization is not described; complete prompts and texts for Experiments 2–5 are not provided.
- No independent human raters or inter-rater reliability; no blinding procedure.
- No statistical analysis or reliability reporting; "No" to statistical significance (page 10, lines 434–441).
- Reproducibility claims conflict with missing details (compute resources, parameters); checklist marks "Yes" for reproducibility (page 10, lines 389–394) despite gaps; compute resources explicitly "No" (page 10, lines 446–455).
- Contradictory checklist entry claiming "theory assumptions and proofs" = Yes (page 9, lines 374–383) though the paper includes no theorems or proofs.
- Use of highly recognizable historical cases without verification that anonymization prevents recognition/memorization by the LLM; no test for case identification.
- Experiments 2–5 only reported as categorical ratings (page 5, lines 203–220) without outputs, evidence, or justification.
- Unsupported technical claims (e.g., GRPO and hallucination rate; page 8, lines 325–328) and uneven citation quality (e.g., [5] non-archival link; possible placeholder citation [9]).
- No operationalized scoring rubric for "depth, consistency, sensitivity"; no coder training or adjudication protocol.
- Ambiguity over whether the "Consistency: Excellent" claim in Experiment 1 (page 4, lines 181–187) is verifiable given missing full inputs and potential hallucinations.

---

### Note · Reviewer_AIRevRelatedWork · 2025-10-06

**Related Work Check**

Please look at your references to confirm they are good.

**Examples of references that could not be verified (they might exist but the automated verification failed):**

- Introduction to Psychoanalysis (Z. Li, Trans.) by Freud, S.

---

### Decision · Program_Chairs · 2025-10-08

**Decision:**

Reject

**Comment:**

Thank you for submitting to Agents4Science 2025! We regret to inform you that your submission has not been accepted. Please see the reviews below for more information.